# PRADA: Prompt-driven Dual Alignment with Actor-Critic Rewards for Aerial-Ground Person Re-Identification

## Abstract

Aerial-Ground person Re-IDentification (AG-ReID) aims to match individuals captured from aerial drones and ground surveillance cameras, posing unique challenges due to severe viewpoint variations, scale discrepancies, and heterogeneous resolutions. To address these challenges, we propose PRADA (Prompt-driven Dual Alignment with Actor-Critic Rewards), a unified framework that combines Prompt-driven Dual Alignment (PDA) and Part-level Actor-Critic Reward Engine (PARE) for robust cross-view representation learning. Specifically, PDA enforces cross-platform identity consistency while preserving intra-platform specificity, and PARE dynamically emphasizes discriminative local features by optimizing complementary classification and confidence rewards under an Actor-Critic paradigm. Additionally, a Cross-Platform Multi-Positive Alignment loss further aligns identity features across aerial and ground domains. Extensive experiments on benchmark AG-ReID datasets demonstrate that PRADA outperforms state-of-the-art methods, validating the effectiveness of integrating prompt-guided alignment with reinforcement-based part-level supervision.The source code can be found here:https://github.com/upgirlnana/PRADA

## 1 Introduction

Aerial-Ground person Re-IDentification (AG-ReID) aims to match individuals captured by aerial drones and ground surveillance cameras. Compared to traditional person ReID He et al. (2021); Li et al. (2023), AG-ReID introduces more severe challenges due to drastic viewpoint variations, scale discrepancies, and heterogeneous image resolutions Nguyen et al. (2023b; 2024); Zhang et al. (2024a). These variations significantly alter the distribution of body parts across views, making it difficult to extract robust features that remain consistent between aerial and ground perspectives.

Most existing Re-ID studies focus primarily on ground-to-ground scenarios, with relatively few investigations addressing aerial-aerial or aerial-ground settings Nguyen et al. (2023b; 2024); Zhang et al. (2024a). The release of specialized datasets such as PRAI-1581 Zhang et al. (2020b) and UAV-Human Li et al. (2021) has catalyzed research on aerial-to-aerial ReID. The AG-ReID.v1 Nguyen et al. (2023b) and AG-ReID.v2 Nguyen et al. (2024) benchmarks further expand capture perspectives to include UAVs, CCTV cameras, and wearable devices, enabling the development of more generalizable algorithms. Additionally, a large-scale synthetic dataset named CARGO Zhang et al. (2024a) and associated methods such as VDT provide a testbed and techniques for mitigating view discrepancies via view-decoupled transformers.

Current research in AG-ReID primarily focuses on three complementary directions, each targeting a specific aspect of the cross-view matching problem.

First, **person attributes** are utilized as auxiliary information to enhance identification performance. The AG-ReID series Nguyen et al. (2023b; 2024) exploit pedestrian attributes to provide stable cues that remain consistent across drastic viewpoint changes, thereby improving cross-view feature discrimination and robustness.

Second, **view-specific variations** are disentangled to mitigate the impact of perspective discrepancies. For instance, VDT Zhang et al. (2024a) separates view-related and view-unrelated features via

hierarchical subtractive separation and an orthogonal loss, while SD-ReID Hu et al. (2025a) employs a generation-based approach using Stable Diffusion to produce view-specific features, explicitly addressing cross-view discrepancies in aerial-ground scenarios.

Third, **prompt- or semantic-based strategies** are employed to facilitate high-level feature learning. VSLA-CLIP Zhang et al. (2024b) leverages prompt-based mechanisms and a Video Set-Level Adapter to transform cross-platform visual alignment into a visual-semantic alignment problem, enhancing semantic feature representation for aerial-ground video ReID.

Together, these approaches, including attribute-guided supervision, view disentanglement, and prompt / semantic-based learning, enhance cross-view consistency and robustness, enabling models to effectively capture both fine-grained local features and high-level semantic representations across heterogeneous camera platforms. Motivated by these observations, we propose **PRADA** (Prompt-driven Dual Alignment with Actor-Critic Rewards) for AG-ReID, which integrates two complementary modules:

- **Prompt-driven Dual Alignment (PDA)**: enforces cross-platform identity consistency, preserves intra-platform compactness, and maintains inter-identity separability;

- **Part-level Actor-Critic Reward Engine (PARE)**: adaptively emphasizes discriminative local features by optimizing part-level classification and confidence rewards.

The main contributions of this work are: First, PRADA provides a unified framework that addresses both cross-platform discrepancies and drastic viewpoint variations. Second, PDA enables robust semantic-level alignment across heterogeneous camera platforms. Third, PARE enhances fine-grained feature representation through adaptive, reward-guided local feature selection. Finally, extensive experiments on multiple AG-ReID benchmarks demonstrate state-of-the-art performance and strong generalization, validating the effectiveness of combining prompt-driven alignment with part-level reinforcement learning.

## 2 RELATED WORK

### 2.1 CROSS-PLATFORM REID

Traditional ReID research has primarily focused on ground-to-ground scenarios Hermans et al. (2017); Li et al. (2023), while recent efforts have shifted toward the more challenging aerial-to-ground setting, enabled by datasets such asPRAI-1581, UAV-Human, AG-ReID, and CARGO Zhang et al. (2020b); Li et al. (2021); Nguyen et al. (2023b). These benchmarks introduce diverse conditions across UAV and ground views, making them critical for evaluating cross-platform ReID models. To address severe view discrepancies, recent methods have explored feature disentanglement and adaptive representation. For example, the View-Decoupled Transformer (VDT) Zhang et al. (2024a) separates view-specific and invariant features via orthogonal constraints, while the Dynamic Token Selective Transformer (DTST) Wang & Pishgar (2024b) selectively retains informative tokens. Further strategies enhance robustness through multi-stream fusion Nguyen et al. (2024; 2025) and generative augmentation Hu et al. (2025a). Overall, aerial-ground ReID research is advancing rapidly through the development of specialized datasets and tailored architectures. Our work builds on these foundations by designing a robust feature alignment framework resilient to viewpoint variation, temporal dynamics, and local distractions.

### 2.2 LEARNING WITH SEMANTIC AND ADAPTIVE GUIDANCE

Prompt-guided and reinforcement learning approaches have recently emerged as promising directions for injecting semantic cues and dynamically selecting informative regions in person re-identification (ReID), especially under challenging aerial-ground conditions.

**Prompt-based Feature Learning** Prompt-based techniques, originally developed for language models, are increasingly applied in vision-language ReID tasks to align visual embeddings with identity-relevant semantics. MP-ReID Zhai et al. (2024) generates explicit (attribute-based) and implicit learnable prompts via ChatGPT and VQA systems to enhance visual alignment. ProFD Cui

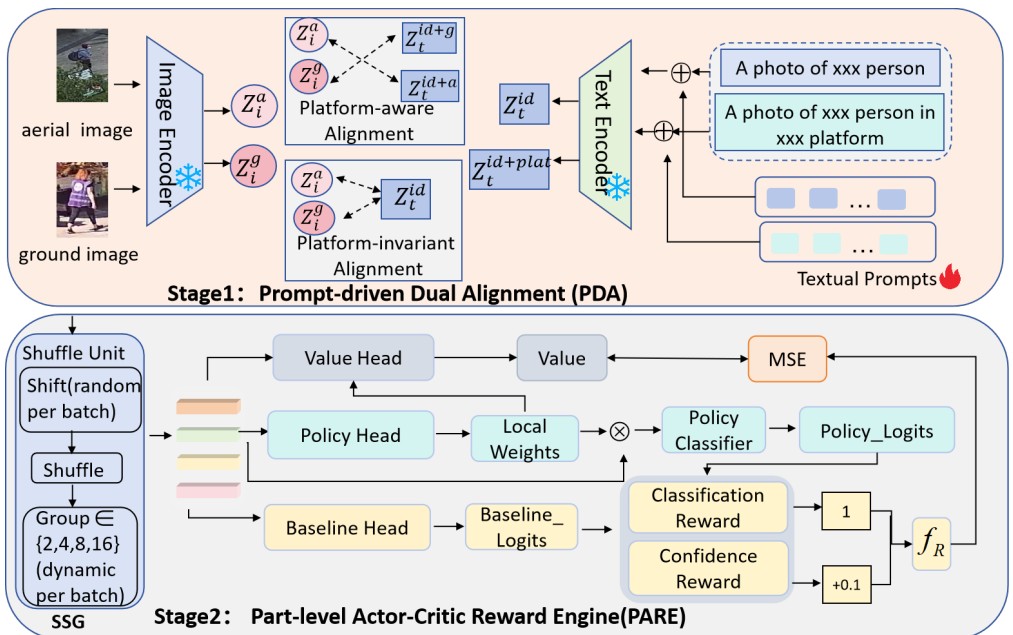

Figure 1: Overview of our proposed framework. The model incorporates three strategies: (1) Prompt-driven dual alignment(PDA) for platform-aware and platform-invariant identity semantics, (2) Dynamic shift- shuffling and grouping for robust local feature learning, and (3) Part-level actor-critic reward engine(PARE) for adaptive local feature refinement.

et al. (2024) uses part-specific prompts guided by segmentation masks, combined with hybrid-attention decoding and self-distillation, to disentangle occluded features while preserving CLIP knowledge. Other approaches, such as $\pi$-VL Lin et al. (2023), leverage human parsing or video-level language prompts to align part-level or cross-modal features, achieving strong performance in ReID and VVI-ReID benchmarks.

**Reinforcement Learning for Feature Selection** Reinforcement learning (RL) has been effective for dynamically selecting informative content, enhancing robustness in occluded or noisy scenarios. Prior works apply RL at multiple levels: region selection to suppress irrelevant image regions Xu (2023), video-frame selection to aggregate discriminative frames for track representations Zhang et al. (2019), and attribute-level selection to filter noisy soft-biometric cues while preserving identity-relevant information Zhang et al. (2020a). Combined with prompt-based methods that provide external semantic guidance, these approaches improve cross-view identity alignment and mitigate occlusions. Our method builds upon these insights, integrating semantic prompts with RL-style adaptive feature selection into a unified framework for aerial-ground ReID.

## 3 METHOD

Our framework is shown in Fig1.

### 3.1 PROMPT-DRIVEN DUAL ALIGNMENT (PDA)

Cross-platform person ReID requires both identity consistency across views and sensitivity to platform-specific differences. To achieve this, we propose **Prompt-driven Dual Alignment (PDA)**, which integrates a shared visual encoder, dual alignment branches, and prompt-guided textual supervision. Specifically, aerial-view and ground-view images are mapped into a latent space using a shared visual encoder, producing feature embeddings $\mathbf{Z}_i^a$ and $\mathbf{Z}_i^g$. We then introduce a dual alignment module with two complementary branches: the *identity-consistent branch*, which learns features $\mathbf{Z}_i^{id}$ that remain stable across platforms, and the *platform-sensitive branch*, which learns

features $\mathbf{Z}_t^{id+plat}$ that capture both identity and platform cues for improved discriminability. To guide this alignment, we design dual-level prompts and encode them with a frozen text encoder. The prompts are defined as: ``A photo of xxx person'' for identity supervision and ``A photo of xxx person in xxx platform'' for joint identity and platform supervision. These textual embeddings serve as semantic anchors, enabling explicit cross-modal alignment.

We formulate the overall loss as a weighted sum of identity-consistent and platform-sensitive alignment losses:

$$\mathcal{L}_{\text{PDA}} = \lambda_{id} \cdot \mathcal{L}_{id} + \lambda_{plat} \cdot \mathcal{L}_{id+plat}, \tag{1}$$

where $\lambda_{id}$ and $\lambda_{plat}$ control the relative contributions. Each loss is instantiated as a contrastive objective between visual and textual embeddings. For example, the identity-consistent loss is

$$\mathcal{L}_{id} = -\log \frac{\exp(\langle \mathbf{Z}_t^{id}, \mathbf{T}^{id} \rangle / \tau)}{\sum_k \exp(\langle \mathbf{Z}_t^{id}, \mathbf{T}_k^{id} \rangle / \tau)}, \tag{2}$$

while the platform-sensitive loss is

$$\mathcal{L}_{id+plat} = -\log \frac{\exp(\langle \mathbf{Z}_t^{id+plat}, \mathbf{T}^{id+plat} \rangle / \tau)}{\sum_k \exp(\langle \mathbf{Z}_t^{id+plat}, \mathbf{T}_k^{id+plat} \rangle / \tau)}, \tag{3}$$

where $\langle \cdot, \cdot \rangle$ denotes cosine similarity, $\tau$ is a temperature parameter, $\mathbf{T}^{id}$ and $\mathbf{T}^{id+plat}$ are the corresponding textual embeddings, and $k$ indexes negative samples.

Overall, PDA disentangles platform-invariant and platform-specific cues under prompt-driven supervision, achieving robust identity alignment across aerial and ground views while preserving platform-aware discrimination. This dual mechanism not only reduces the domain gap between aerial and ground views, but also avoids losing fine-grained discriminative cues, ultimately enhancing both cross- and intra-platform ReID performance.

## 3.2 Dynamic Shift and Shuffle Group Strategy(SSG)

To improve feature robustness in aerial-ground ReID, we propose a *dynamic shift and shuffle group* mechanism. Given input features $\mathbf{F} \in \mathbb{R}^{B \times N \times D}$, we first apply a random cyclic **shift** along the token dimension, followed by **shuffle grouping** into $g$ segments sampled from $\mathcal{G} = \{2, 4, 8, 16\}$ with slightly unbalanced sizes. This dynamic perturbation enforces *group-invariant* representations, mitigates overfitting to fixed token orders under large viewpoint changes, and acts as a lightweight feature-space augmentation, enhancing cross-platform discriminability and generalization.

## 3.3 PARE: Part-level Actor-Critic Reward Engine

To enhance fine-grained feature selection, we design a **Part-level Actor-Critic Reward Engine (PARE)** that integrates baseline supervision with policy-driven exploration. Specifically, we employ a single backbone with three heads: a *baseline head* for classification, a *policy head* for learning adaptive feature weights, and a *value head* for estimating expected rewards. The policy head operates at the part-level, where each local feature branch independently interacts with the environment and produces enhanced logits.

To provide effective supervision, we introduce two complementary *part-level rewards*. Let $\mathbf{z}_{base}^{(p)}$ and $\mathbf{z}_{pol}^{(p)}$ denote the baseline and policy-enhanced logits of the $p$-th local feature ($p = 1, \ldots, 4$), and $y$ the ground-truth identity. The first is a **classification reward**, which evaluates whether the policy-enhanced classifier correctly predicts the ground-truth identity:

$$R_{\text{cls}}^{(p)} = \begin{cases} 1, & \arg\max(\mathbf{z}_{pol}^{(p)}) = y, \\ 0, & \text{otherwise}, \end{cases} \tag{4}$$

which provides binary feedback to drive the policy towards emphasizing discriminative regions.

Beyond correctness, we further design a **confidence reward** to refine this supervision by encouraging the policy to improve its predictive confidence relative to the baseline:

$$R_{\text{conf}}^{(p)} = \max \left( 0, \ \pi_{pol}(y) - \pi_{base}(y) \right), \tag{5}$$

where $\pi_{pol}(y)$ and $\pi_{base}(y)$ denote the softmax probabilities of the ground-truth identity under policy and baseline logits, respectively.

Finally, we unify both signals into a parametric form:

$$R^{(p)} = \alpha \, R_{\text{cls}}^{(p)} + \beta \, R_{\text{conf}}^{(p)}, \tag{6}$$

where $\alpha$ and $\beta$ balance discrete correctness with soft confidence gains. The critic network evaluates the long-term value of these rewards, enabling stable policy optimization via an Actor-Critic framework.

Beyond local supervision, we further incorporate a **Cross-Platform Multi-Positive Alignment (CP-MPA) Loss** to explicitly enforce *intra-platform compactness*, *cross-platform consistency*, and *inter-identity separability*. For a sample $x_i$ with identity $y_i$ and platform $p_i$, we construct multiple positives $\mathcal{P}(i)$ from both intra- and cross-platform instances sharing the same identity. The CP-MPA loss is formulated as:

$$\mathcal{L}_{pal} = -\frac{1}{B} \sum_i \log \frac{\sum_{p=1}^P \exp(\text{sim}(\mathbf{x}_i, \mathbf{t}_{i,p})/\tau)}{\sum_{j \neq i} \sum_{p=1}^P \exp(\text{sim}(\mathbf{x}_i, \mathbf{t}_{j,p})/\tau) + \sum_{p=1}^P \exp(\text{sim}(\mathbf{x}_i, \mathbf{t}_{i,p})/\tau)}, \tag{7}$$

where $\text{sim}(\cdot, \cdot)$ denotes cosine similarity and $\tau$ is a temperature parameter. By constructing multiple positive pairs across both intra- and inter-platform samples of the same identity, CP-MPA aligns identity features while preserving discriminability.

Together, PARE dynamically guides the model to attend to the most informative local features and ensures robust identity alignment across heterogeneous platforms. The final loss in the PARE stage combines baseline, policy, and value losses:

$$\mathcal{L}_{PARE} = \lambda_1 \mathcal{L}_{reward} + \lambda_2 \mathcal{L}_{tri} + \lambda_3 \mathcal{L}_{cls} + \lambda_4 \mathcal{L}_{pal} \tag{8}$$

$$\mathcal{L}_{\text{reward}} = \frac{1}{|P|} \sum_{p \in P} \mathcal{L}_{\text{reward}}^{(p)} \quad \text{and} \quad \mathcal{L}_{\text{reward}}^{(p)} = \frac{1}{4} \sum_{i=1}^4 \left( v_i^{(p)} - R_i^{(p)} \right)^2. \tag{9}$$

## 4 EXPERIMENTS

### 4.1 DATASETS AND EVALUATION PROTOCOLS

We evaluate our method on three aerial-ground person re-identification (AG-ReID) benchmarks. **AG-ReID.v1** Nguyen et al. (2023b) contains 21,983 images of 388 identities with 15 attributes from aerial and ground cameras, supporting two protocols (A→G, G→A). **CARGO** Zhang et al. (2024a) is a large-scale synthetic dataset with 108,563 images of 5,000 identities captured by five aerial and eight ground cameras. **AG-ReID.v2** Nguyen et al. (2024), an extension of AG-ReID.v1, adds a wearable-device (W) view to aerial (A) and ground (G), providing 100,502 images of 1,615 identities and enabling diverse protocols (A↔W, G↔W).

### 4.2 METRICS

We adopt mean Average Precision (mAP) Zheng et al. (2015) and Cumulative Matching Characteristic (CMC) Moon & Phillips (2001) at Rank-1 as the primary evaluation metrics.

### 4.3 IMPLEMENTATION DETAILS

Our method is implemented in PyTorch and trained on four NVIDIA RTX 3090 GPUs with CLIP-Base-16 as the backbone. We use a batch size of 64 (16 identities, 4 images each). Training consists of two stages: in Stage 1, only learnable text tokens are optimized using Adam with an initial learning rate of $3.5 \times 10^{-4}$ under cosine decay; in Stage 2, we linearly increase the learning rate from $3.5 \times 10^{-6}$ to $3.5 \times 10^{-4}$ over 10 epochs, then decay it by 0.1. A warm-up of 10 epochs is applied with the learning rate growing from $5 \times 10^{-7}$ to $5 \times 10^{-6}$. Data augmentation includes random cropping, color jittering, and random erasing, with all images resized to $256 \times 128$. The model is fine-tuned for 120 epochs with Adam ($3.5 \times 10^{-4}$).

## 4.4 PERFORMANCE COMPARISON

We compare our proposed method with existing methods on three AG-ReID benchmarks in Tab 1,Tab2 and Tab3. The experimental results across three distinct datasets—AG-ReID.v1, CARGO, and AG-ReID.v2—demonstrate the superior performance of our proposed method. As shown in the provided tables, our model consistently surpasses existing state-of-the-art methods by a significant margin. On the AG-ReID.v1 dataset, our method achieves a Rank-1 accuracy of 86.9% and an mAP of 79.9% for the A→G task, and 89.3% (Rank-1) and 82.9% (mAP) for the G→A task, establishing new benchmarks. Similarly, on the CARGO dataset, our model leads in most tasks, particularly with a Rank-1 of 73.1% and an mAP of 66.7% for the ALL task, highlighting its robustness. Furthermore, our method's exceptional performance on the more challenging AG-ReID.v2 dataset, with highest scores in all four tasks (e.g., Rank-1 of 92.3% and mAP of 87.9%for A→W), validates its effectiveness in handling complex cross-domain scenarios. The consistent outperformance across diverse datasets and tasks underscores the effectiveness of our approach and its potential to set a new standard in the field of person re-identification.

Table 1: Performance comparison on CARGO. The best and second-best results are highlighted in **bold** and underline.

| Method | ALL | | A↔G | | G↔G | | A↔A | |
|---|---|---|---|---|---|---|---|---|
| | **Rank-1** | **mAP** | **Rank-1** | **mAP** | **Rank-1** | **mAP** | **Rank-1** | **mAP** |
| SBS He et al. (2023) | 50.3 | 43.1 | 31.3 | 29.0 | 72.3 | 63.0 | 67.5 | 49.7 |
| BoT Luo et al. (2019) | 54.8 | 46.5 | 36.3 | 32.6 | 77.7 | 66.5 | 65.0 | 49.8 |
| MGN Wang et al. (2018) | 54.8 | 49.1 | 31.9 | 33.5 | 83.9 | 71.1 | 65.0 | 53.0 |
| VV Kumar et al. (2020) | 45.8 | 38.8 | 31.3 | 29.0 | 72.3 | 63.0 | 67.5 | 49.7 |
| AGW Ye et al. (2021) | 60.3 | 53.4 | 43.6 | 40.9 | 81.3 | 71.7 | 67.5 | 56.5 |
| ViT Dosovitskiy et al. (2020) | 61.5 | 53.5 | 43.1 | 40.1 | 82.1 | 71.3 | 80.0 | 64.5 |
| VDT Zhang et al. (2024a) | 64.1 | 55.2 | 48.1 | 42.8 | 82.1 | 71.6 | **82.5** | **66.8** |
| PRADA | **73.1** | **66.7** | **64.4** | **58.8** | **84.8** | **77.8** | 65.0 | 51.3 |

Table 2: Performance comparison on AG-ReID.v2. The best and second-best results are highlighted in **bold** and underline.

| Method | A→C | | A→W | | C→A | | W→A | |
|---|---|---|---|---|---|---|---|---|
| | **Rank-1** | **mAP** | **Rank-1** | **mAP** | **Rank-1** | **mAP** | **Rank-1** | **mAP** |
| SwinLiu et al. (2021) | 68.8 | 57.7 | 68.5 | 56.2 | 68.8 | 57.7 | 64.4 | 53.9 |
| HRNet-18(Wang et al., 2020) | 75.2 | 65.1 | 76.3 | 66.2 | 76.3 | 66.2 | 76.3 | 66.2 |
| SwinV2Liu et al. (2022) | 76.4 | 66.1 | 80.1 | 69.1 | 77.1 | 62.1 | 74.5 | 65.6 |
| MGN(R50)Wang et al. (2018) | 82.1 | 70.2 | 88.1 | 78.7 | 84.2 | 72.4 | 84.1 | 73.7 |
| BoT(R50)Luo et al. (2019) | 80.7 | 71.5 | 86.1 | 75.9 | 79.5 | 69.7 | 82.7 | 72.4 |
| BoT(R50)+Attributes | 81.4 | 72.2 | 86.7 | 76.7 | 80.2 | 70.4 | 83.3 | 73.1 |
| SBS(R50)He et al. (2023) | 81.9 | 72.0 | 88.1 | 78.9 | 84.1 | 73.9 | 84.7 | 75.0 |
| SBS(R50)+Attributes | 82.6 | 72.7 | 88.7 | 79.6 | 84.8 | 74.6 | 85.3 | 75.7 |
| ViTDosovitskiy et al. (2020) | 85.4 | 77.0 | 89.8 | 80.5 | 84.7 | 75.9 | 88.6 | 80.1 |
| V2E(ViT)Nguyen et al. (2024) | 88.8 | 80.7 | **93.6** | 84.9 | 87.9 | 78.5 | 88.6 | 80.1 |
| PRADA | **88.8** | **83.6** | 92.3 | **87.9** | **88.7** | **83.6** | **90.6** | **85.1** |

## 5 ABLATION STUDIES

To thoroughly evaluate the contribution of each proposed module, we conduct ablation experiments on the Aerial ReID benchmark. Specifically, we start from a baseline model without any of our modules and progressively integrate them to observe performance improvements. The Aerial results and visualization analysis are in the appendix. The baseline model only utilizes the global class tokens from the visual encoders.

### 5.1 EFFECTS OF KEY MODULES.

**Effect of SSG and PARC.** Model A serves as the baseline, achieving a mAP of 59.8% and a Rank-1 accuracy of 67.5% on the PRAI-1581 dataset, and 73.0% mAP and 70.5% Rank-1 on the UAVHu-

Table 3: Performance comparison with existing methods on AG-ReID.v1. The best performance is shown in **bold**.

| Method | A→G | | | G→A | | |
|---|---|---|---|---|---|---|
| | **Rank-1** | **mAP** | **mINP** | **Rank-1** | **mAP** | **mINP** |
| OSNet Zhou et al. (2021) | 72.6 | 58.3 | – | 74.2 | 61.0 | – |
| BoT Luo et al. (2019) | 70.0 | 55.5 | – | 71.2 | 58.8 | – |
| SBS He et al. (2023) | 73.5 | 59.8 | – | 73.7 | 62.3 | – |
| ViTDosovitskiy et al. (2020) | 81.3 | 72.4 | – | 82.6 | 73.4 | – |
| FusionReID Wang et al. (2025) | 80.4 | 71.4 | – | 82.4 | 74.2 | – |
| CLIP-ReID Li et al. (2023) | 79.4 | 70.6 | – | 84.2 | 73.1 | – |
| Explain Nguyen et al. (2023a) | 81.5 | 72.6 | – | 82.9 | 73.4 | – |
| VDT Zhang et al. (2024a) | 83.0 | 74.1 | 50.3 | 84.6 | 76.3 | 49.5 |
| DTST Wang & Pishgar (2024a) | 83.5 | 74.5 | 49.9 | 84.7 | 76.1 | 50.0 |
| SD-ReID Hu et al. (2025b) | 85.2 | 75.4 | 51.4 | 86.0 | 77.0 | 50.4 |
| PRADA | **86.9** | **79.9** | | **89.3** | **82.9** | |

Table 4: Ablation study on CARGO and AG-ReID.v2

| Method | CARGO | | AG-ReID.v2 | | | | | | | |
|---|---|---|---|---|---|---|---|---|---|---|
| | ALL | | A→C | | A→W | | C→A | | W→-A | |
| | Rank-1 | mAP | Rank-1 | mAP | Rank-1 | mAP | Rank-1 | mAP | Rank-1 | mAP |
| w/o Eq.(7) | 73.1 | 66.7 | 87.4 | 81.3 | 91.7 | 86.3 | 87.2 | 80.9 | 90.0 | 82.9 |
| w/ Eq.(7) | 73.4 | 67.2 | 88.3 | 82.8 | 91.5 | 86.5 | 87.4 | 81.6 | 89.8 | 83.7 |

man dataset. The introduction of the SSG module in Model B leads to a slight performance improvement, achieving 62.8% mAP and 70.1% Rank-1 accuracy on PRAI-1581, as well as 74.9% mAP and 75.4% Rank-1 on UAVHuman. These results suggest that the dynamic SSG enhances robustness by enforcing group-invariant features, mitigating overfitting to token order, and providing lightweight feature-space augmentation, thereby yielding more discriminative representations. Model C incorporates all three modules (SSG, $PARE_{R_{cls}}$, $PARE_{R_{conf}}$), resulting in a mAP of 63.0% and a Rank-1 accuracy of 70.2% on PRAI-1581, and 75.4% mAP and 76.2% Rank-1 on UAVHuman. The best performance is observed in Model C, which utilizes all modules, achieving a mAP of 63.0% and a Rank-1 accuracy of 70.5% on PRAI-1581, and 75.8% mAP and 76.5% Rank-1 on UAVHuman. These results indicate that the integration of the SSG, and PARC modules contributes to the robustness and effectiveness of the model.

**Effect of Eq.(7).** We conduct this experiment on CARGO and AG-ReID.v2 since both datasets contain rich cross-view samples with large appearance variations. As shown in Table 4, incorporating Eq.(7) consistently improves the mean Average Precision (mAP). While the Rank-1 accuracy shows only marginal changes, mAP exhibits more stable and evident gains, e.g., +1.5 on the A→C split and +0.8 on W→A. Since mAP reflects the overall ranking quality across all positive samples rather than a single top-1 match, this improvement indicates that Eq.(7) encourages the model to retrieve a larger portion of true positives at higher ranks. We attribute this to the nature of Eq.(7), which simultaneously pulls multiple positive samples closer in the embedding space. Such design reduces intra-class variance across different views and conditions, leading to more robust retrieval in multi-view scenarios.

# 6 CONCLUSION

In this paper,we introduced a novel framework named PRADA for AG-ReID. By integrating the *Prompt-driven Dual Alignment (PDA)* module with the *Part-level Actor-Critic Reward Engine (PARE)*, PRADA simultaneously achieves cross-platform consistency and discriminative part-level representation learning. The proposed classification and confidence rewards provide stable reinforcement signals, while the Cross-Platform Multi-Positive Alignment loss explicitly enforces robust identity alignment. Comprehensive experiments validate the superiority of PRADA over state-of-the-art approaches. In future work, we plan to extend PRADA to broader multi-modal and cross-domain recognition tasks, further exploring its scalability and adaptability.

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
