

Figure 1: Ablation study on PRAI-1581 and UAVHuman.

# PRADA: Prompt-driven Dual Alignment with Actor-Critic Rewards for Aerial-Ground Person Re-Identification

## Appendices

We organize the appendix as follows:

- **Appendix A**: Ablation study results on Aerial datasets.
- **Appendix B**: Extended experimental visualizations

## A Ablation Study on PRAI-1581 and UAVHuman

This bar chart presents an ablation study comparing the performance of different modules on two datasets: PRAI-1581 and UAVHuman. The study evaluates performance using two metrics: Rank-1 accuracy and mean Average Precision (mAP).The specific discussion is in the paper.

## B Visualization Analysis.

**Feature Distributions.**

As shown in Fig. 2, we visualize the feature distributions of the baseline model and our method using t-SNE. The baseline model tends to produce scattered clusters with ambiguous boundaries, where samples of the same identity are dispersed and partially overlapped with other classes. In contrast, our method, with SSG and PARE, yields more compact and clearly separable clusters, indicating that intra-class compactness and inter-class separability are significantly improved. These visualizations fully validate the effectiveness of our proposed modules in improving feature discrimination.

**Rank list Comparison.**

As illustrated in Fig. 3, we present a qualitative comparison of rank list results between the baseline model and our method. For each query, the odd rows (first and third) display the retrieval results of the baseline model, while the even rows (second and fourth) show the results of our method. It can be observed that the baseline model often retrieves visually similar but incorrect candidates, highlighted by red boxes. n contrast, with the integration of SSG and PARE, our model yields rank lists that are progressively more accurate and discriminative. which demonstrates its stronger discriminative capability and robustness against appearance variations and background interference.

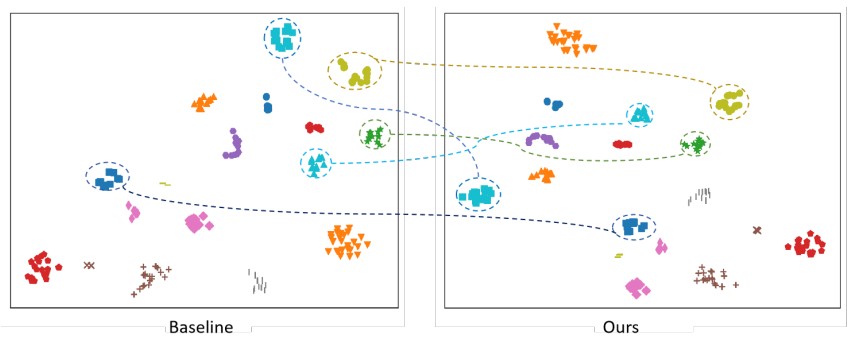

Figure 2: Feature distributions with t-SNE. Different colors refer to different IDs.

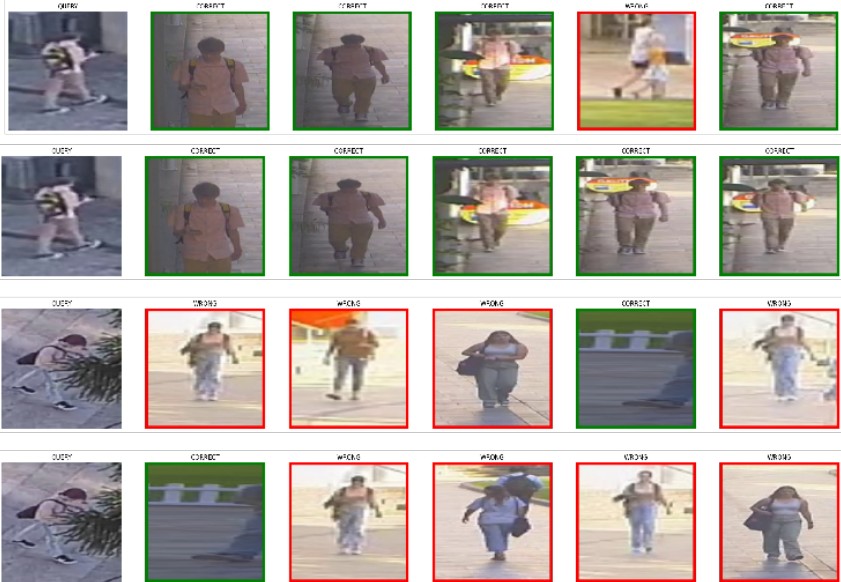

Figure 3: Rank list comparison of different models.Odd rows (the first and third) correspond to the baseline model, while even rows (the second and fourth) show the results of our method.