# OpenReview forum: "PRADA: Prompt-driven Dual Alignment with Actor-Critic Rewards for Aerial-Ground Person Re-Identification"
_ICLR.cc/2026/Conference — ICLR 2026 Conference Desk Rejected Submission_

### Official Review · Reviewer_NtF3 · 2025-10-25

**Soundness:** 2
**Presentation:** 2
**Contribution:** 3
**Rating:** 4
**Confidence:** 4

**Summary:**

This paper addresses challenges in AG-ReID, such as viewpoint variations and scale discrepancies, by introducing a Prompt-driven Dual Alignment (PDA) to mitigate cross-platform viewpoint changes and employing a Part-level Actor-Critic Reward Engine (PARE) with a Shuffle Group Strategy (SSG) to enhance discriminative local features. The effectiveness of the proposed approach is validated on publicly available datasets, such as CARGO.

**Strengths:**

1. To address cross-platform discrepancies, this paper introduces abundant semantic information as guidance, enabling the model to pull identity-related features closer while distinguishing platform-specific features.
2. By employing a reinforcement learning design with an actor-critic reward engine at the part level, this paper dynamically enhances discriminative local features and suppresses similar features.

**Weaknesses:**

About PDA:
1. Regarding the effectiveness of PDA, there is a lack of relevant ablation experiments to prove it.

About SSG&PARE:
1. Lines 205-206: Missing explanation and experimental evidence for p=4
2. The network structure of PARE is not introduced.
3. Regarding the statement 'dynamically emphasizes discriminative local features' corresponding images need to be added to demonstrate that attention is indeed focused on local features.

Others：
1. The content in Figure 1 is inconsistent with or unclear compared to the description below.
2. Lines 230-232: The introduction to PAL loss is insufficient, and the meanings of the symbols are not explained.
3. Regarding the t-SNE part, only the olive green part and blue triangle part sections appear noticeably clustered, while no significant differences are observed in other parts, and since there is only one set of comparisons, the persuasiveness is weak.

**Questions:**

1. SSG performs cyclic shifts on tokens. Does this process damage the synmantic of human body thus the pedestrian's ID information?
2. SSG is random, and Group is random. Does this mean that the meaning represented by part is random, such as tokens of different body details mixed within the same part? Could this lead to suboptimal performance of PARE?
3. The training of Actor-Critic rewards is relatively difficult. How does this paper achieve stable training, and what is its computational complexity?

---

### Official Review · Reviewer_Jy6D · 2025-10-28

**Soundness:** 2
**Presentation:** 2
**Contribution:** 2
**Rating:** 2
**Confidence:** 4

**Summary:**

This paper proposed PRADA (Prompt-driven Dual Alignment with Actor-Critic Rewards), a unified framework that combines Prompt-driven Dual Alignment (PDA) and Part-level Actor-Critic Reward Engine (PARE) for robust cross-view representation learning. The paper presents PRADA (Prompt-driven Dual Alignment with Actor-Critic Rewards), a unified framework that integrates Prompt-driven Dual Alignment (PDA) and Part-level Actor-Critic Reward Engine (PARE) for robust cross-view representation learning. The authors have conducted a series of comparative experiments on multiple ReID datasets, and the results demonstrate the effectiveness of the proposed method. Moreover, the ablation studies explore the impacts of SSG, PARC, and CP-MPA.

**Strengths:**

(1) Originality: This paper introduces a reinforcement learning strategy in the AG-ReID field, proposing a Part-level Actor-Critic Reward Engine (PARE), which provides a new perspective for solving cross-platform recognition issues. The introduction of this method reflects the authors' innovative attempts based on existing research.

(2) Effectiveness: The effectiveness of the method presented in this paper has been validated through comparative experiments with other methods.

**Weaknesses:**

(1) The paper lacks a comparative analysis of the principles between PDA and the method in VSLA-CLIP, which transforms the cross-platform visual alignment problem into a visual-semantic alignment problem.

(2) The explanation of the reinforcement learning method PARE (PART-LEVEL ACTOR-CRITIC REWARD ENGINE) is unclear.

(3) The comparative experiments are insufficient: the paper uses CLIP's image and text encoders to extract features, but lacks comparison with CLIP-ReID in the experiments.

(4) The ablation studies are inadequate: ablations are only conducted on PRAI-1581 and UAVHuman datasets, without experiments on other datasets (e.g., CARGO, AGReID.v1, AGReID.v2), and there is no ablation study on the PDA module.

(5) The specific values of hyperparameters are not provided, and there is a lack of parameter sensitivity analysis.

(6) The writing is unclear, especially in the experimental section. It is recommended to use a table in Section 5.1 to present the performance of each experiment.

(7) Formatting issue: the tables appear to be too large and exceed the page margins.

(8) Important: The paper includes a non-anonymous GitHub link. According to the ICLR policy—“ICLR 2026 is double blind, which means that all submitted papers should be anonymous. Any paper where author identity is revealed in either the main text or the supplementary material will be desk rejected.” I am seriously concerned that this submission may violate that rule. I would appreciate it if the AC (or others) could conduct a further check. For clarity, I have not taken this issue into account in my review.

**Questions:**

(1) What is the difference between the PDA module in this paper and the approach in VSLA-CLIP that transforms the cross-platform visual alignment problem into a visual-semantic alignment problem?

(2) The explanation of the reinforcement learning method PARE (PART-LEVEL ACTOR-CRITIC REWARD ENGINE) is unclear. For example, what is in Figure 1? What is the relationship between Stage 1 and Stage 2? It seems that Stage 1 is about learning prompts, but Stage 2 does not clarify what these prompts are used for. What does the double arrow between Value and MSE mean?

(3) Since this paper uses CLIP's image and text encoders to extract features, why are no comparisons made with methods that also use CLIP as the backbone in the comparative experiments? For example, why is there no comparison with CLIP-ReID?

(4) Why are ablation studies only conducted on the PRAI-1581 and UAVHuman datasets? What about other datasets such as CARGO, AGReID.v1, and AGReID.v2? Also, why is there no ablation study on the PDA module?

(5) What are the specific values of the hyperparameters? Why is there no parameter sensitivity analysis?

---

### Official Review · Reviewer_B7Pp · 2025-11-01

**Soundness:** 2
**Presentation:** 2
**Contribution:** 2
**Rating:** 4
**Confidence:** 4

**Summary:**

In this paper, the  Prompt-driven Dual Alignment with Actor-Critic Rewards (PRADA) method is proposed for aerial-ground person re-identification (AG-ReID). It aims to provide guided alignment for cross-view identity retrieval under viewpoint and scale variations using prompt-based dual alignment (PDA) and a part-level actor-critic reward engine (PARE). PDA enforces cross-platform identity consistency while preserving intra-platform specificity using CLIP-style text embeddings. PARE enhances fine-grained local feature learning via reinforcement learning, using classification and confidence rewards to emphasize discriminative parts. A cross-platform multi-positive alignment loss is also proposed to align identity features across aerial and ground domains. Experiments on AG-ReID.v1/v2 and CARGO data show that PRADA can surpasses SOTA baselines. Ablations showing that both PDA and PARE contribute complementary benefits.

**Strengths:**

+ Overall, the main components of PRADA are described in enough detail to understand the paper.
+ The paper focuses on aerial-ground person ReID a newer and challenging task for UAV and surveillance applications.
+ The authors effectively link prompt-driven alignment across camera domains, and reinforcement-based feature refinement. They integrate PDA with an PARE from visual-semantic alignment and adaptive part selection.
+ Empirical results show that PRADA can outperform on SOTA aerial-ground baselines: AG-ReID.v1/v2 and CARGO. Ablations are show the benefits of combining  PDA and PARE.
+ Supplementary material has additional information on ablations in aerial datasets, and more visual results that help support the paper.

**Weaknesses:**

- The methodology section (especially Section 3.3) leaves a lot of questions. It does not provide a clear justification for design choices, e.g., architecture, losses, choice of rewards, weighting parameters, is weak. Several components of PRADA are undefined or poorly motivated.
- The prompt design is limited to simple text templates, without exploring adaptive or learnable prompt tuning
- The main novelty of the work is mostly the Actor-Critic Reward module, but it is not clearly isolated. The ablations (Fig 1 in supplementary, Sec 5.1 in main paper) shows its contribution paired with the loss in Eq. 7, making it difficult to assess its standalone benefit.
- This paper should also contain an experimental analysis of time and memory complexity.  Computational overhead from multiple branches and reward updates is not analyzed, raising concerns about scalability.
- The paper has formatting inconsistencies (dense tables, misplaced figures, excessive manual spacing) that affect readability and presentation quality.
- Their code is not made available, so there is a concern that the results in this paper would be difficult for a reader to reproduce.

**Questions:**

See my comments in weaknesses. Also, In Table 1, why is the PRADA performance lower for the last column? I don’t see any interpretation for this result.

**Details Of Ethics Concerns:**

None.

---

### Note · Program_Chairs · 2026-01-17
**Submission Desk Rejected by Program Chairs**

The following references in this submission do not refer to real documents and/or have major errors in bibliographic information:

 Hyeonjoon Moon and P Jonathon Phillips. Performance measures of verification and identification in biometric systems. Proceedings of the 16th International Conference on Pattern Recognition (ICPR), 4:4-7, 2001.